# A Medical Equipment Lifecycle Framework to Improve Healthcare Policy and Sustainability

Bruce Mang [1,*] , YeonJae Oh [2] , Chabelly Bonilla [3] and Jennifer Orth [3,*,†]

1    College of Natural Sciences, University of Texas at Austin, Austin, TX 78712, USA
2    Faculty of Health Sciences, Queen's University, Kingston, ON K7L 3N6, Canada; rose.oh@queensu.ca
3    College of Arts and Sciences, Seton Hall University, South Orange, NJ 07079, USA; chabelly.bonilla@gmail.com
*    Correspondence: brucemang@utexas.edu (B.M.); leigh011@gmail.com (J.O.)
†    International Rotary Fellowship of Healthcare Professionals: Re-MERGE.

**Abstract:** The healthcare sector is struggling to become more environmentally friendly compared to other industries, evidently seen by the contribution to global emissions. These struggles have elicited some research on sustainable methods regarding the lifecycle of medical devices. Indeed, the World Health Organization (WHO) encourages the reuse of equipment and ethical donations, namely for the sake of the environment and sustainable global health. However, there is a lack of synthesis–multiple greener alternatives to the current healthcare system are developing without a connection to each other, hindering an increase in sustainability. Thus, there is a lack of global organization and standardization in medical equipment lifecycles. Inspired by the findings and guidelines of the Safe and Sustainable Medical Equipment Supply Subgroup (SASMES) of the International Rotary Fellowship of Healthcare Professionals, we created the Re-processing Medical Equipment: Rotarian Research Group for the Environment (Re-MERGE) to expand on these challenges. Re-MERGE follows the life cycle of medical devices in the United States of America through its initial stages of classification and various regulatory pathways, the middle stage of post-market requirements, and the end stage of disposal or donation and reprocessing. Our findings indicate that current medical device end-stages are inefficient, damaging to the environment, and burdensome to donation recipients; however, existing processes can provide improvements to medical device end-stage methods by drastically reducing environmental damage, improving healthcare globally, and increasing sustainability in the field. We identify that more research is needed to connect the implications of different medical device end stages. Additionally, we encourage the findings to be implemented to create more sustainable, effective methods of medical device disposal, donation, and reprocessing.

**Keywords:** medical devices; medical device lifecycle; environmental impacts; medical device ethics; healthcare sustainability

## 1. Introduction

The global impact of the healthcare sector ranges between one and five percent of the total global impact depending on the indicator [1]. The global healthcare sector is the fifth largest carbon emitter on earth, producing 4.4 percent of net carbon emissions [1]. Specifically, the US healthcare sector is responsible for 25% of global healthcare emissions [2]. A major contributing factor to this is healthcare facilities, which combine to produce more than five million tons of waste each year [3]. Unfortunately, the industry of medical devices is falling behind in terms of sustainability due to heavy regulations and rigorous policies [4]. As defined by the United Nations (UN) World Commission on Environment and Development, sustainability entails "development that meets the needs of the present without compromising the ability of future generations to meet their own needs". These issues have implications reaching into global health and ethics. Although there have been improvements in medical device sustainability, there are still knowledge gaps in medical

device life cycles. Specifically, attention to the current end-stages of medical equipment is important to help reduce medical waste and protect the environment. Additionally, improving aspects of medical device end stages can advance global health sustainability and ethics. We provide an analytical approach to medical equipment donation, disposal, and reprocessing through literature searches and interviews. Finally, the information presented in sections other than Donations mainly pertains to the U.S. healthcare system and high-income countries. Our findings and discussion are detailed below and serve as an informative perspective regarding the medical equipment lifecycle and considerations on actions for their end stages.

## 2. Background: Rotary International and Terminology Definitions

### 2.1. Rotary International

In the fall of 2021, the United Nations concluded its 26th Climate Change Conference in Glasgow discussing issues such as implementing the Paris Agreement, adapting to achieving their goals, and emphasizing the importance of communication in planning and acting. These issues served as a scientific supplement to the Safe and Sustainable Medical Equipment Supply (SASMES) "Good Practice Guide", which was generated by the International Rotary Fellowship of Healthcare Professionals as an effort to spread awareness of greener ways to reprocess donated medical equipment and provide information for the group at large (This Fellowship operates in compliance with Rotary International's policies, but it is not an agency of, or controlled by Rotary International) [5]. This paper is structured chronologically throughout the lifecycle of a medical device. Under the section "End stage", there are subsections detailing and discussing the implications of different aspects of the end stage of a medical device.

### 2.2. Refocusing Definitions

There are numerous definitions involved in the medical donation process that need clarification for the intended use of the terminology of this paper:

Remanufacturing/Repurposing vs. Reprocessing vs. Refurbishing

Remanufacturing and repurposing a device can be used interchangeably [6]. This process involves decommissioning a device outside of the original manufacturer's intention and significantly changes the device's original performance or safety specifications. For example, a CT scanner that is decommissioned to be used on animals is considered a remanufactured or repurposed device.

Reprocessing involves the sterilization, disinfecting, and possible remanufacturing of a piece of equipment back to the United States of America (USA) Food and Drug Administration (FDA) and original manufacturer standards for reuse. This process involves the largest organizations, such as hospitals which typically reprocess equipment for reuse [6]. For example, a blood pressure sleeve that is sent back to be re-serviced and sterilized for reuse is considered reprocessed.

Refurbishing indicates the relinquishment of an equipment's title by a hospital to a third-party vendor. In the USA, the device is no longer subject to FDA and manufacturer regulations. These companies dispose of or donate the equipment to places of need. In the European Union (EU), refurbishing and reprocessing a device are used interchangeably [6].

## 3. Early Stages of the Medical Device Life Cycle: Device Classification and Regulatory Pathways

In the United States, the life cycle of a medical device usually begins with concepts and planning to target market demands and evaluate device risks [7]. A prototype is designed to determine its practicality and feasibility in the market using various methods, such as 3-D printing, powder bed fusions, and computer numerical control machining [7].

The proposed device must also be under FDA-specified classes to advance. The FDA classifies about 1700 different generic types of devices grouped into 16 medical specialties

referred to as panels [8]. These devices are then sorted into classes depending on the level of control necessary to ensure the safety of the device. Class I devices pose a minimal threat to consumers and are mostly exempt from the regulatory process [8]. They are non-invasive and involve a few moving parts, such as bandages and wheelchairs. Most devices fall under Class II, which can be best described as devices that pose a moderate to high risk to patients, such as ventilators and CGMs [8]. Class III devices prolong human life and are often invasive [8]. As such, they present major risks when defective and are heavily regulated by lawmakers and the FDA. Examples include stents, pacemakers, and artificial hips. The class determines the types of pre-market submissions and applications required for FDA clearance for the market, which also has an impact on the length of the process. For example, about 74% of Class I devices are exempt from premarket notification processes [9]. Regardless of class, all devices are subject to General Controls, the baseline requirements of the Food, Drug, and Cosmetic Act [9]. Notably, the intended and indicated purpose of a device plays a large role in the classification as this can lead to extra approval processes due to specificity [8]. FDA runs a classification site and panels that can be used to find the class of a specific device. After a device is formally classified, manufacturers can move onto regulatory pathways for FDA approval.

The next stage in a medical device's life cycle is approval, in which the device undergoes a specific regulatory pathway via the FDA [7]. Multiple pathways are taken depending on a device's classification and purpose [7]. The pre-market approval (PMA) pathway suits new, unclassified, or Class III devices [9]. This pathway is the most rigorous, costing $94 million per piece of equipment on average and taking months to complete [10]. Approximately 5% of medical devices are brought to the market each year via PMA [10].

Another pathway is the 510k pathway which validates devices that already have similar devices on the market [11]. Moreover, 510k is the most popular pathway, entailing 90% of new medical devices, averaging $31 million in comparison [10]. Applicants support the substantial equivalence (SE) claims by comparing novel devices to existing equipment. By FDA definition, SE implies that the legally marketed device has the same intended use and technology. Some examples of 510k's include dialysis machines, x-ray machines, and fetal monitors.

Manufacturers also use four lesser-known pathways for specific instances. The De Novo pathway allows low-risk devices, such as class I devices, to avoid the PMA pathway even if a similar device is not already on the USA market. This path is rapidly growing due to its cost savings and logical approach [11]. Humanitarian device exemption (HDE) is a pathway that encourages treatments for rare diseases [11]. While HDE is fast and cheap, it is subject to certain profit restrictions due to its purpose [11]. Custom device exemption is the least utilized pathway. This pathway is used typically for providers seeking a custom device that will not be mass-manufactured [11]. Only five of these devices are to be made per year [11]. Finally, the product development protocol (PDP) pathway allows the sponsor to reach an early agreement with the FDA on demonstrating the safety of the new device (FDA). At the manufacturer's pace, the device obtains PMA when the PDP is declared as completed by the FDA [11]. This method is advantageous when the technology of that medical device is already well-established on the market [11]. After a device has gone through these pathways, it moves onto the middle stage of its life cycle.

## 4. Middle Stages of the Medical Device Life Cycle: Post-market Requirements and Maintenance

A device is market-ready and will operate during the middle stages. These stages include FDA post-market requirements. A device may require tracking systems, malfunction reports, and proper manufacturing documents [12]. Guidance documents that explain the use of the device to the public and the FDA are sometimes required [12]. In addition, the FDA partnered with medical device stakeholders to create the National Evaluation System for Health Technology (NEST) [13]. NEST gathers evidence throughout the life cycle of medical devices and applies analytics to data tailored to the innovation cycles of

medical devices. NEST synthesizes these data from many places such as clinical registries, electronic health records, and billing claims [13]. This information could help healthcare workers and patients decide on the right treatment by balancing safety and innovation. Furthermore, post-market surveillance is conducted for each device. Any adverse reports result in recalls or safety warning amendments, posted annually by the FDA.

Additionally, maintenance is an important component of a medical device's functioning life. Equipment maintenance must pass various qualifications for use within the FDA, Medicare, and the non-profits themselves. The WHO has a medical equipment maintenance program under which they categorize maintenance under two categories: Inspection and Preventive Maintenance (IPM) and corrective maintenance (CM) [14]. IPM includes all pre-booked and arranged visits and activities, preventing any equipment failures, thus cleaning, recalibrating, updating, etc. medical equipment [14]. CM encompasses all issues that need correcting, including technical issues and extending the life of the equipment [14]. Through this program, the essence of running proficient maintenance will be constant, regardless of if the maintenance is needed in an urban area of a HIC or a rural setting in an LMIC [14]. Through maintenance, a healthcare facility can ensure the efficacy of all equipment and devices [14]. Due to the varied ways that organizations maintain equipment, below are diverse ways organizations upkeep their equipment.

## 5. End Stage: Disposal, Donation, and Reprocessing

A device's life cycle can then proceed down either the path of disposal, donation, or reprocessing. Currently, incineration of medical waste is the predominant form of disposal. As of 2000, approximately 60% of regulated medical waste (RMW) has been incinerated, 37% has been sterilized by steam, and 4–5% are treated via other methods [15]. After disinfection, RMW is disposed of in places regulated by the Environmental Protection Agency (EPA): landfills, surface impoundment units, waste piles, land treatment units, and injection wells [16]. Contamination-proof landfills are the most common method of disposal storage [16]. While the disposal of medical devices ends their life cycle, the equipment can continue to provide value via proper reprocessing and donation.

## 6. Disposal: Environmental Impacts and Solutions

Medical waste disposal currently poses a threat to environmental sustainability globally. Medical waste can be classified into two categories: general waste and special waste [15]. While general waste is not regulated or defined as dangerous, special waste is deemed potentially hazardous and requires special handling [15]. Regulated medical waste (RMW) includes special wastes deemed potentially health hazardous specifically by the Medical Waste Tracking Act (MWTA), such as chemical, infectious, and radioactive waste [15]. Incineration is the most popular method because of its waste-volume reduction, putrefaction prevention, waste-heat recovery routine, and overall efficiency in disposing of contaminated substances [15]. While effective and affordable, this method releases harmful dioxins, furans, and mercury due to the large amounts of plastic in medical waste–a catalyst to potentially irresponsible policies. Evidently, due to unclear sorting and instructions, quantities of non-infectious devices are also incinerated which is environmentally and financially harmful. On average, the USA disposes of 3.5 million metric tons of medical waste a year, with each ton costing $790 [15]. Hospitals in the USA generate an average of 139 tons of on-site incineration waste [15]. For off-site disposal, another 2334 tons worth of infectious, chemical, and radioactive waste are shipped off to disposal facilities every year [15]. Medical device supply chains are responsible for 80% of the emissions, such as greenhouse gases, from healthcare sources in the USA [17]. Holistically, the USA healthcare sector contributes 9–10% of greenhouse gases [3]. In addition to contamination, the United Nations predicts that 75% of pandemic-related medical waste such as masks are expected to end up in landfills or the oceans [18]. This is problematic because improper segregation of medical or biomedical waste could contaminate groundwater sources and elicit a domino

effect of infections on animals and humans. However, this does not need to be the case as not all waste needs to be shipped to off-site disposal locations or incinerated.

There are potential solutions to alleviate the impacts caused by the lack of sustainability in the healthcare system, such as sustainable disinfection methods and accurate waste classification. Instead of incineration disposal, autoclaving and microwave disinfection can be used for certain devices [15]. The EPA passed emission regulations on incinerators in 1997, making them less affordable and pushing hospitals to seek alternatives [19]. After this upstream intervention, healthcare centers adopted cleaner strides toward disinfection. The Cleveland Clinic only uses steam sterilization in place of incineration since 2009 [19]. Future financial incentives and policies can be developed to facilitate this movement further. In terms of waste classification, NEST articulates that simply accurate classification of waste can still decrease the cost of disposal [13]. By implementing clear and strict policies in conjunction with classification training, health centers can work together to minimize environmental damage and energy waste from medical disposal [13].

Additionally, software visibility, where each medical practice setting uses a database to track expiration dates, helps to prevent and reduce waste. Ashlea Souffrou, President of SxanPro, works with hospitals in sustainability protocols. Souffrou asserts that out of the forty-two states she visited in the last decade, almost all hospitals lacked visibility in their medical equipment inventory stock. Most hospitals, regardless of size, do not know their stock and inventory in detail and are inclined to stock-pile devices instead of organizing their supplies [20]. The lack of communication between hospital departments and the absence of software systems that organize a hospital's inventory are the reasons why even the largest health systems do not have visibility of expiration dates [20]. Lack of knowledge about important dates for medical devices causes hospitals to stockpile equipment and renders the circulation of devices stagnant, which leads to 5–7% of total spending on supplies going expired or unused [20]. Expired equipment becomes medical waste, which eliminates the chance of being transferred, sold, or donated [20]. Medical device reprocessing must be completed in a timely manner, as the effectiveness is dependent on the time of contact between internal and external forces. Souffrou thus recommends the use of software visibility, which allows hospitals to mitigate the amount of waste generated, increase the management of medical equipment inventories, and effectively allocate money spent on devices. Using SxanPro's savings calculator, about 30k to 100k could be saved every year depending on hospital size [20]. Queen Elizabeth Hospital in the United Kingdom has been involved in software visibility. Queen Elizabeth Hospital uses a management system called Backtraq to track details of specific equipment in their inventories [21]. Backtraq allows the tracking of equipment [21]. With a database of medical device codes, medical engineers can track the usage of the device [21]. This serves to automatically manage maintenance recall or disposal and ensures that a marginal number of resources go to waste [21]. Utilizing resources for management enforces a 'green' mentality that could lead to an increase in other environmentally friendly hospital policies.

## 7. Reprocessing: Environmental Benefits, Obstacles, and Recycling

Medical device reprocessing as an alternative to disposal has recently seen a surge as associations such as the Association of Medical Device Reprocessors (AMDR) have almost doubled in size annually [6]. With this method, a device can continue its life cycle as long as reprocessing standards meet manufacturer and FDA regulations. Medical device reprocessing is also environmentally beneficial and prevents the loss of over 600,000 disability-adjusted life years (DALY) annually [3]. AMDR quantifies that in 2018 alone, reprocessing diverted 15 million pounds of medical waste from landfills and saved healthcare institutions an estimated $544 million [18]. In particular, hospitals that have medical devices reprocessed by regulated reprocessors have removed over 7100 tons of waste that would have otherwise harmed the environment in landfills [18]. These techniques, part of the circular economy of medical equipment, have been shown to halve the emission of $CO_2$ and other greenhouse gases as well as 30% less usage of manufacturing resources [18]. In

turn, this would allow hospitals to develop without compromising the environment for the future. However, there are cultural and regulatory obstacles to reprocessing (Table 1).

**Table 1.** Average New and Reprocessed Prices of Common Medical Equipment [22].

| Device Type | New Price (USD) | Reprocessed Price (USD) |
|---|---|---|
| Cardiac catheters | 280 | 60 |
| Orthopedic surgical blades | 30 | 14 |
| Deep vein thrombosis compression sleeves | 125 | 11 |
| Laparoscopic instruments | 1240 | 250 |
| Tourniquet cuff | 25 | 12 |
| Saw blade | 40 | 18 |
| Ultrasound catheters | 2900 | 1400 |
| Laparoscopic shears | 120 | 55 |
| Cardiac stabilizers | 900 | 380 |
| Pulse oximetry sensor | 10–20 | 6–10 |

The potential and importance of such reusable equipment come with a price. Medical equipment pricing has varied estimates that affect reprocessing. For instance, a portable unit in the radiology field is approximately $1000 USD, with most large units ranging from $20,000–$50,000 [23]. According to Harold Rodriguez, the director of the Radiology department at Mount Sinai–Beth Israel Hospital in New York City, a "bidding war" or auction, is commenced for reprocessed equipment. Once the bidding war is over, the hospital will confirm the highest bidder and best offer [23]. The claimant of the unit would also pay to remove it from the hospital of origin [23]. The price for removal can range from $10,000 to $100,000 [23]. The Emergency Care Research Institute provides guidelines on how to prepare a reprocessed piece of equipment for donation and reusable equipment, and the PriceGuide Database for accurate pricing on reprocessed equipment.

Medical equipment can be sold to a third-party reprocessor for reprocessing due to profitability. The costs of purchasing and selling medical equipment for either reprocessing or donation depend on the equipment, manufacturer, and hospital [23]. Many hospitals undergo a bidding war to obtain and sell medical equipment for reprocessing. Some healthcare establishments may also contract all reprocessing duties to third parties. For instance, Mount Sinai–Beth Israel in New York City, needs at least three parties interested in the equipment for the bidding to advance [23]. In May of 2021 the radiology department at Mount Sinai–Beth Israel was auctioning off a CT scanner [23]. An animal clinic was interested in the scanner, and they were most likely the victor in that specific bidding war [23]. If sold to them, the original manufacturer would have to reprocess the scanner for animal use and update the software for improved use [23].

### 7.1. Single-Use-Devices (SUD)

A prominent issue surrounding the reprocessing of medical equipment is the number of SUDs in the market. SUDs constitute a significant number of medical devices and equipment, as the development of plastic polymers has made SUDs "booming" since the 1970s [24]. Hence, addressing SUDs is a key factor in sustainable medical equipment practices. SUDs belonging to medical device classes I and II can be reprocessed [25]. However, most healthcare professionals are unaware that governmental regulations allow the reprocessing of SUDs and latch on to the preconceived notion that it is unsafe to reprocess SUDs [26]. Daniel J. Vukelich, President and CEO of AMDR explains that the fear of cross-contamination roots back to the HIV pandemic when a consumable culture propagated the notion that disposable equipment should be preferred over reusable equipment [6]. On top of environmental damage, Vukelich furthers that economically, SUDs are less cost-efficient. It is estimated that 40–50% of the original disposable list price is saved when using reusable equipment such as surgical drapes [6]. As a matter of fact, Neurosurgeons at Toronto Western Hospital were able to reduce disposable usage by 30%

and save $570,000 USD [27]. Beneficially, Sally McDonald from the Charles Perkins Centre and School of Pharmacy writes that institutions that are prepared to reuse equipment can quickly respond to demand surges [17]. They are less impacted by supply chain disruptions, manufacturing shortages, price shocks, or trade [17]. Importantly, the SARS-CoV-2 pandemic has shed light on the potential and importance of reusable equipment, and the public is starting to pay attention.

*7.2. Recycling*

In conjunction with reprocessing and reusable equipment, improvements in recycling are also essential to increase sustainability in medical equipment lifecycles. Approximately three million tons of non-infectious medical plastics–widely utilized due to convenience, safety, lightweight, and usability–can be recycled in the USA, Europe, and Asia combined [19]. Such recycling includes single-use plastics, which constitute at least 20% of USA hospital medical waste [26]. Nevertheless, there is an absence of a widespread, effective medical recycling system because of a lack of knowledge. For instance, an over-classification of biohazard medical waste in hospitals leads to missed recycling or reprocessing opportunities. Medical plastics that could be recycled are dumped into red biohazard garbage bags. Indeed, a survey conducted across four USA Mayo Clinic locations reported that 48% lacked knowledge about recycling, 39% 'never' or 'only sometimes' recycle, and 57% could not tell which items in operating rooms could be recycled [26]. Academic literature has found that a substantial amount of hospital waste could be disposed of at municipal landfills but instead, is processed as infectious waste costing $0.79 per kg as opposed to $0.12 per kg [15]. Contrary to the current state, only 15% of total healthcare waste is biohazardous [28]. There are two solutions to this problem. First, is the design of medical devices. Even if the design phase requires a lot of money, there must be an emphasis placed on reprocessing and recycling potential. Indeed, 80% of sustainable decisions are made in the initial design process of medical devices [29]. Second, creating an effective medical recycling system starts with training healthcare workers in medical waste disposal. There are actions to be made at every stage of disposal (pre-disinfection, disinfection, and post-disinfection) that can alleviate current environmental impacts. Before disposal occurs, infectious waste can be sorted with more precision and accuracy. A lack of standardization makes effective waste sorting difficult for healthcare workers who err on the side of caution, which causes unnecessary infectious waste generation [15]. Accordingly, with a training intervention that educated medical staff on the proper disposal or handling of medical equipment, hospitals were able to save 10% of their costs in sharp waste disposals [26]. Waste-sorting standardization could push hospitals to dispose of equipment in more energy-efficient, cleaner methods. The combination of training and innovation could accelerate the path to an eco-friendly health industry. As such, future advancements in the industry include growing sustainable packaging coalitions and organizations such as the Healthcare Plastics Recycling Council (HPRC) to create a unified and organized way of increasing sustainability in the medical sector [19]. These changes are necessary to provide a path for medical development while ensuring that cost and environmental damage remain minimal.

Finally, medical equipment can be decommissioned for donation when it is not reprocessed, or newer technology comes out with better results [23]. While reprocessing is an environmentally friendly method for the healthcare industry in high-income countries, donations are a viable alternative to help low-middle-income countries (LMICs) with better health outcomes and sustainability.

## 8. Donation: Inherency and Follow-Up Care of Medical Devices

A major humanitarian alternative to environmentally damaging disposal is medical device donation. Medical device donations have the capability to circumvent the negative impacts of current popular disposal methods while improving healthcare in LMICs. However, currently, donations face roadblocks: lack of staff training in recipient facili-

ties, missing functional parts for devices, and communication standards regarding the donation processes. This disproportionately affects low-income countries which often receive equipment without proper instructions, functioning parts, or support. Over 80% of medical equipment in low-income and middle-income countries (LMIC) is donated from high-income countries (HIC) [30]. However, there are no global regulations on these donations as they are typically executed by an agreement or preset guidelines between the host government/institution and the donor [30]. This is problematic, as medical equipment to be donated is manufactured for use in a HIC infrastructure such as the voltage of wall plugs, which is unsuitable for LMICs without proper resources and preparation [30]. In the early 21st century, these organizations and the WHO have made strides to ensure that donations are sustainable, beneficial, and responsible in the long term.

An issue of *International Health* analyzed 88 reports from media and journals describing 53 donation initiatives: disaster donations, long-term donation programs, or one-off donations [17]. Some examples include the Chinese government donating medical equipment to the Steve Biko Academic Hospital in South Africa or non-government organizations (NGOs) such as the American Medical Resources Foundation donating medical devices to the ADENI Hospital of Quetzaltenango [17]. However, a considerable amount of medical device donation initiatives resulted in a burden to the recipient countries in terms of storage and disposal costs due to violations of WHO guidelines (Table 1) [3,18]. An estimated 40–70% of medical equipment was not functioning or lacked staff training for operation, maintenance, or disposal [17]. One finding noted large amounts of donations in quick succession which overwhelmed local services. In this case, equipment would expire and possibly result in environmental pollution without proper disposal, causing more harm than good [17].

However, there are cases that demonstrate proper donation guideline compliance, such as the Johnson and Johnson donation of mebendazole for the control of soil-transmitted helminthiasis. The program included proper application training, proposed medicine distribution, monitoring, and evaluation to ensure the endemic areas were prioritized. Paperwork was arranged beforehand and any issues with shipments or delays were evaluated. Similarly, other cases show good compliance with WHO guidelines. In 2012, diagnostic devices were donated to Takaya health center in the Democratic Republic of Congo [17]. The donation was chosen based on the physicians with experience in the area and available resources were taken into consideration. Likewise, the devices were low-cost, durable, and solar-powered recharging systems were included. Efforts to minimize importation complications were in place and the devices were evaluated quarterly [17]. The steps taken mirrored WHO guidelines for donations and yielded a positive effect on the receiving country. This is a step towards sustainability because global health measures such as diagnostics reduce downstream healthcare interventions. In addition to being more environmentally harmful, these interventions reduce the quality of life for people.

Objectives that are clearly shown to be crucial include communication between the host and recipient, support for maintenance, and upstream measures to allow the functionality of the medical device in the recipient country. Not all projects follow these procedures which can lead to complications. For instance, in 2012 and 2013, healthcare facilities in Uganda received medical equipment with poor WHO guideline compliance [17]. Donations appeared to be 'dumped' at the facilities and were often provided just to meet corporate responsibility targets. The devices were either broken immediately or soon after arrival. They also could not be repaired due to a lack of spare parts and training of any kind. Likewise, the equipment often did not even come with all the required parts to function. Any given instructions were not provided in commonly spoken languages in Uganda, and electrical needs were not met [17]. As a result, the conclusion that was reached indicated there was currently too large of a burden on countries in need in terms of storage, sorting, and disposal in combination with a lack of training, parts, and functionality to be largely beneficial. In cases such as these, medical device donations become potentially dangerous due to the lack of maintenance in terms of parts and service. Although there is

no governing body to dictate medical device donations, compliance with medical device donation guidelines can greatly increase the efficacy of donations that would otherwise cause damage to patients and the environment. Table 2 shows responses to medical donations from various countries that aligned with WHO guidelines.

**Table 2.** Percentage of medicine donations compliant with guideline items [17].

| Guideline Item | Number of Donations that Met Guideline | Number that Reported on Compliance | Percentage (%) Compliant with Guideline Item |
|---|---|---|---|
| 1.1. Is the donation based on an expressed need from the recipient? | 25 | 29 | 86.2 |
| 1.2 Is the donation relevant to the disease pattern of the recipient country? | 27 | 28 | 96.4 |
| 1.3 Were the donation quantities agreed upon between the donor and recipient? | 12 | 17 | 70.6 |
| 2.1 Are the donated medicines or their generic equivalents approved for use in the recipient country? | 7 | 10 | 70.0 |
| 2.3. If 'no' to question 2.2, was the donation specifically requested by the recipient? | 1 | 1 | 100.0 |
| 3.1 Is the presentation, strength, and formulation of the donation similar to those of medicines commonly used in the recipient country? | 0 | 3 | 0.0 |
| 4.1 Was the donation obtained from a quality-ensured source? | 26 | 28 | 92.9 |
| 4.2 Does the donation comply with the quality standards of the donor country? | 0 | 1 | 0.0 |
| 4.3 Does the donation comply with the quality standards of the recipient country? | 1 | 4 | 25.0 |
| 4.4 Was the WHO Certification Scheme on the Quality of Pharmaceutical Products Moving in International Commerce used? | 0 | 0 | 0.0 |
| 5.1 Was the donation free from returned/recycled medications, or free samples were given to health professionals? | 20 | 21 | 95.2 |
| 6.1 Do the donated quantities match the recipient countries' consumption needs before they are expired? | 2 | 5 | 40.0 |
| 7.1 Are the medicines labeled in a language easily understood by healthcare workers in the recipient country? | 1 | 3 | 33.3 |
| 9.1 Was the donation packed in accordance with international shipping standards, accompanied by a detailed packing list, and not mixed with other supplies (unless shipped as kits with predetermined contents)? | 3 | 6 | 50.0 |
| 10.1 Were medicines sent with the prior consent of the recipient? | 24 | 26 | 92.3 |
| 12.1 Were the associated costs of transport, storage, port clearance, handling, disposal, etc. paid for by the donor (unless agreed upon with the recipient in advance)? | 27 | 27 | 100.0 |

Similarly, donated equipment needs continued maintenance to extend the lifetime of the reprocessed equipment. This may include preliminary checks before they are accepted to be donated. Some third-party companies in charge of donations from a hospital often provide follow-up care and routine checks on said equipment. Brother's Brother Foundation is a charity organization in the USA that distributes medical equipment to LMICs across the globe [31]. They follow the Healthcare, Infrastructure, Disaster Response, and Education Initiative (H.I.D.E) which has regularly scheduled relief shipments across the globe [31]. The foundation prioritizes electricity failure as part of its maintenance, so they install solar power backup systems in the emergency of a power outage and temperature-controlled storage to help with the efficacy of certain projects [31].

The MedSurplus Alliance (MSA), a medical surplus recovery organization (MSRO), acquires any new and used medical supplies, equipment, and pharmaceutical products from USA healthcare institutions, manufacturers, and distributors [32]. MSROs then inspect the donated products to ensure it meets their standards and make them accessible to low-income healthcare providers via direct shipments [32]. They also provide follow-up care such as medical volunteer teams, local non-profit organizations, and training [32]. Many hospitals, clinics, and manufacturers donate to MSROs. The equipment is then evaluated by engineers and technicians to validate that the donated products are up to MSA's standards and code of conduct [33]. Under the code of conduct, any expired, unsanitary, or "unfixable" products will not be accepted for donation and end up disposed of or sold to reprocessing companies [33]. Donating products is an effective way to prevent medical equipment waste that usually ends up in landfills, thus lessening our carbon footprint.

Additionally, a centralized database that connects all parties involved in medical donations is integral to the efficacy of device donations to LMICs. Medical facilities or non-profit organizations that help run downstream or upstream interventions in LMICs should not have to excessively search for available equipment by contacting multiple organizations separately. Rather, they should be able to access a database where they can match with different organizations, charities, medical device engineers, and institutions to obtain a relatively convenient way of obtaining reprocessed medical equipment. There are many organizations that already carry out donations such as Brother's Brother and MSA.

The aforementioned stages entail the entire life cycle of a medical device. Figure 1 showcases this pathway.

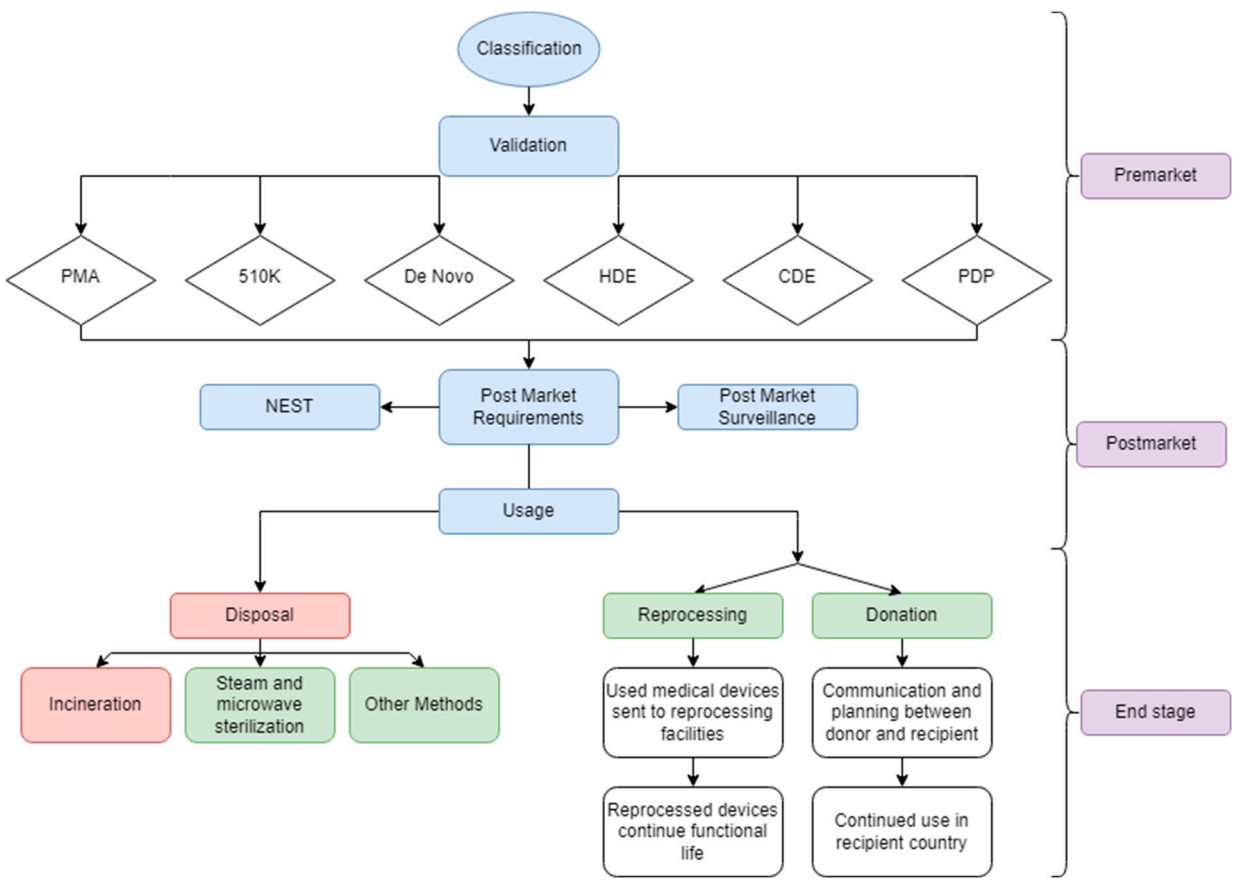

**Figure 1.** Pathway of medical equipment.

## 9. Bioethics of Medical Device Donations

Bioethics is a field of study that converges medicine and the humanities, creating a study of thought that incorporates autonomy, justice, beneficence, and non-maleficence

back into medicine. When facing the issue of how medical equipment donations affect people, many bioethical concerns arise including what the standard of care for equipment may be and if non-maleficence is the main obligation when donating equipment. Dr. Bryan Pilkington, professor of bioethics at Seton Hall University and the School of Health and Medical Sciences at Hackensack Meridian Health offered his expertise.

Pilkington corroborates that donated medical equipment ought to still meet the same standard of care when donated to countries regardless of HIC or LMIC status [33]. No nation deserves better equipment than the other based on the status of living in a HIC or LMIC. Patients should have the right to efficient and operational equipment. Aside from the equipment itself, donated medical equipment ought to have follow-up care such as including a generator when donating equipment that requires having a technician on both the recipient and donor sides to discuss any repairs. Moreover, non-maleficence, to do no harm can affect the amount of medical equipment a donor supplies. This can be attributed to ensuring the effectiveness of equipment. Donating a large quantity of poor equipment is worse than donating a few good pieces of equipment. The donor ought to ensure the proper standards and regulations of reprocessed medical equipment are met to protect the patient's well-being on any side of the world. Both considerations are good examples of what to discuss in the planning process prior to medical equipment donation. Bioethics ought to be a part of the conversation in matters regarding medical equipment donation in order to ensure the best treatment for patients.

## 10. Discussion: Complications and Future Directions

Our framework aims to increase sustainability and healthcare quality through medical device lifecycle policies and management. However, there are often tradeoffs in terms of sustainability and efficacy. Additionally, there are issues with how to incentivize these changes. Some of these obstacles lie within the end stages of medical devices.

While we propose alternatives to disposal, such as microwaves, autoclaves, and steam sterilization, these methods are more costly. Incineration is still the cheapest method of RMW disposal. Our suggestions on classification reforms should alleviate these RMW disposal pressures, but there is a need for incentives regarding these procedures. There is also a general lack of data on specific regulated medical wastes within the past 10 years.

We mention the monetary benefits of reprocessing, recycling, and effective waste and inventory classification. While it is evident that there is a reduction in waste due to increased equipment turnover and decreased device backlog, it is unclear where the savings produced by software visibility programs are spent. These benefits cannot be guaranteed to be invested in improving sustainability. Hence, there is a need to incentivize the healthcare system to invest in a sustainable future with the money they saved from software visibility programs for a better balance between sustainability and patient care. We also found while researching that many different institutions had contradicting data about pricing, recycling, donating, etc.

WHO donation guidelines center on improving global health while maintaining sustainable practices. However, this would disincentivize companies from donating, as proper donation guidelines require additional effort and investment. For example, providing instructions, training, maintenance, and ensuring that the equipment is fully functional is certainly a less attractive option than disposal. There are also ethical issues that arise with donations. Is the equipment currently meeting the best standard of care? Is it going to be donated to a specific population? Is follow-up care and knowledge provided to ensure efficacy and less harm to the patients? Are the same policies of reprocessed medical equipment for HIC accessible to LMICs?

Further research is required to clarify these potential obstacles to our proposed framework. Though our research has implications for global health and sustainability, we recognize that a majority of our policy discussion thus far mainly pertains to the U.S. healthcare system. Future steps include research and frameworks for other countries' medical device lifecycle regulations. Further discussion within the industry would also help the

discrepancy between data of reprocessed devices. Finally, donations and bioethics must always be considered in addition to pursuing sustainability and healthcare improvements.

## 11. Conclusions

Our findings in regard to medical device lifecycles indicate that a more systemized approach in conjunction with changes in methods towards the end stages would be beneficial to the global healthcare field. Reprocessing and recycling medical equipment can lead to a more sustainable and effective healthcare system. Additionally, designing medical equipment for ease of reprocessing can reduce the load of currently environmentally harmful disposal. Even for infectious, non-reusable wastes, alternatives for disposal such as microwaves and autoclaves are more sustainable and eco-friendlier than incineration. In conjunction with effective medical waste sorting, these methods can aid the future of global standardization of medical equipment. Finally, standardizing medical device donation guidelines and follow-up care would greatly improve the LMIC's healthcare resources and experience of accepting donated reprocessed equipment.

While the healthcare industry is always innovating and discovering more efficient, sustainable practices, many of these changes are not realized. Our discussion of sustainability and effective healthcare trade-offs explains some potential obstacles. Additionally, we recognize that incentives and proper investment of saved money must occur to fully realize this framework. This includes our aforementioned solutions to medical device impacts, which can impact healthcare globally. Many of the replaced medical equipment qualify for alternatives to disposal that are not realized due to culture, policy, or inefficient practices. However, these alternatives can help bridge the gap between low-and-high income countries' access to healthcare, lessen the environmental impacts of disposing of medical equipment, and focus more on the bioethical issues at hand and donation processes. We hope our perspective and synthesis of medical device life cycle implications can tie together these issues and solutions to illustrate the changes needed for a better future in healthcare. Whilst our analysis of the lifecycle of medical equipment is primarily focused on the United States of America, we urge our global colleagues to discuss and implement policies on the messages presented in this perspective.

**Author Contributions:** Conceptualization, J.O.; Methodology, B.M.; Validation, B.M. and J.O.; Formal Analysis, B.M., Y.O. and C.B.; Investigation, B.M., Y.O. and C.B.; Resources, B.M., Y.O. and C.B.; Data Curation, B.M., Y.O. and C.B.; Writing—Original Draft Preparation, B.M., Y.O. and C.B.; Writing— Review & Editing, B.M., Y.O. and C.B.; Visualization, B.M., Y.O. and C.B.; Supervision, B.M. and J.O.; Project Administration, B.M. and J.O.; Funding Acquisition, B.M. All authors have read and agreed to the published version of the manuscript.

**Funding:** This research received no external funding.

**Data Availability Statement:** No new data were created and analyzed in this study. Data sharing is not applicable to this perspective.

**Acknowledgments:** We would like to thank the support from the board of the International Rotary Fellowship of Healthcare Professionals, especially John Philip, Chairman of the International Rotary Fellowship of Healthcare Professionals, and Rainer Moosdorf, whose constant support, feedback, and encouragement have been much appreciated. We are grateful to be given the opportunity to conduct this research along the path of SASMES guidelines. We would also like to thank our amazing interviewees, Dan Vukelich of AMDR, Ashlea Souffrou of SxanPro, Harold Rodriguez of the Radiology Department at Mount Sinai—Beth Israel Hospital, and Bryan Pilkington, Bioethics professor at Seton Hall University for providing their time and their insight to our research.

**Conflicts of Interest:** The authors declare no conflict of interest.

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
