# Peer review of "A Medical Equipment Lifecycle Framework to Improve Healthcare Policy and Sustainability"

_challenges, doi:10.3390/challe14020021_

Round 1

Reviewer 1 Report

The article emphasizes one of the most difficult issues to address with a view to the implementation of sustainable processes that seriously address the management of biosanitary waste. The global approach and the consideration of how much of the equipment that is currently discarded can be reused has seemed very valid to me.

Author Response

Thank you so much for your feedback! We truly appreciate it. We have taken your critiques into consideration, specifically to the conclusion. We added a discussion section toward the end of our paper to discuss some impediments, possible future directions, and possible outcomes. We also took the time to define sustainability and how it applies to our framework throughout the paper. We also changed our paper to a framework rather than a conceptualization to fit our argument more. Overall, we really appreciate your observations on our paper. Please let us know what you think of our changes. 

Reviewer 2 Report

It is a good idea to review the literature for healthcare sustainability. Imagining a globally accepted set of standards might be somewhat over-ambitious aim, and the content of your ms is tightly focused on USA, which has a model for healthcare which differs from that in many other countries. It might be argued that the healthcare system you describe in available for  richer Americans. Much of the detail on the USA administrative system works is therefore not of great relevance to a global readership. The scope of references is somewhat limited and there are important gaps. You appear to assume that sustainability embraces both maximum environmental protection and the highest quality of medical care, but in practice actions designed to improve both often involve trade-offs. How would that relate to  a systems framework, which you support?  Defining sustainability is problematical and contested, and you need to explain what you take the term to mean, defining it tightly enough to provide an explicit basis for your work. It is not clear to me how your section on donation is clearly linked to sustainability, and you fail to explain how this will aid achieving sustainability (rather than provide better healthcare for some people). You see advantages in actions which are both environmentally friendly and also enhance efficiency and thus cost saving. There is an extensive literature on the relationship between sustainability and efficiency, and the associated effects of rebound and backfire. Whether of not efficiency advances sustainability depends on what the money saved is spent. Thus, an assumption that  efficiency is always supportive of sustainability is problematic. In relation to plastic waste, recent research has focused on the possibly critical issue of the possible impacts of nano plastics ingestion on health, and the need to take urgent actions to reduce nano plastic ingestion to avoid this possible risk. This work identifies the use of plastic packaging as both a great help in preventing cross infection, and also a very important source of nano plastic. Recycling plastic is complicated where items are formed of components each manufactured using differing polymers.

Some minor points: 

lines 175-6 this sentence seems out of place and the intended meaning is unclear

179 repetition of an idea covered in an earlier section

261 and 275-6 'bidding war' is a vernacular expression, best to stick to technical terms

291 neurosurgeons doesn't need a capital

293 identifies an author by name, but most are not named: standardise ofr all

297-8 'public attention' needs either explanation as to its relevance here, or deleted

The value of Table 1 to an international readership is not clear to me

'Fig 2' (there is no Fig 1 in the ms I read) similarly is not relevant in this way

I think that if these points are taken and relevant modifications made, there is a value in having this detail available for a readership, and it can be useful in the process of providing a framework for future work, but I am not clear how such a framework would merit being called conceptual

Author Response

Thank you very much for your feedback! There are many things we did not notice that you brought up. We have attempted to correct our paper based on your comments. In the introduction, we have clarified what sustainability means to us and tied the definition more clearly throughout the paper. We have also clarified that most sections of the paper pertain to the U.S. healthcare system. You raise a good point that the framework is not conceptual, so we changed the title of our paper because it suits our perspective more. We moved figure 1 (formerly mistakenly labeled figure 2) to a more relevant place within the paper. Your argument about deciphering the differences between efficiency and sustainability was integral to restructuring and adding information to our paper. Since we also noticed insufficient discussion about tradeoffs and complications, we added a discussion section to address these and future directions. Nevertheless, we are still unsure how to phrase the idea of biohazard plastic bags, as our voice that the usage of those bags could be reduced by proper worker training does not seem strong or coherent enough. We have addressed all minor points as well. 

Regarding your comments on the donation section, this section does pertain to improving healthcare. However, it also increases sustainability because adhering to WHO guidelines would reduce improperly disposed of medical equipment. We hope that the mention of reducing improper disposal of waste clarifies that. Finally, regarding your point on author name identification, we used the names of those we interviewed while not including any author names. Should we remove all names? 

Let us know if these changes are what you are looking for; we really appreciate your insight and efforts!

Reviewer 3 Report

The paper is very interesting from the point of view of sustainable economy/waste management, although I think it would be good to draw a parallel with the influence of medical equipment manufacturers and the adaptation of equipment and products (which is partially mentioned in the text) to the needs of medical workers, as well as the adaptation of equipment and products for easier maintenance and repairs , durability, recycling and disposal when they expire (usability).

I think it would be good to give some concrete examples and concrete figures as examples (either devices or consumables) and a (pie) chart showing the quantities of produced and recycled or refurbished materials and devices. The text only approach to writing an article like this makes it poorer.

It would be interesting if somewhere there is a warning that the export of obsolete equipment to third countries can represent a dangerous practice of exporting electronic or medical waste, in the case when materials and devices whose expiration date is close are being knowingly exported.

On the technical side, the literature is written in the wrong font, the table and image are out of the text formatting.

Overall, the article is interesting and useful, but it would be good to show some more specific data/figures for specific types of products/waste.

Author Response

Thank you very much for your feedback! We have recognized that the paper would be enriched with more consent with concrete examples and figures. Thus, we have included one figure (Table 2) that explains the average new and reprocessed prices of common medical devices. This would support the fact that reprocessing reduces operational costs. We could not find the quantities of produced, recycled, or reprocessed medical devices as there is no data about those exact numbers; in fact, when we interviewed Dan Vukelich and asked if there was information, he said that data about quantities are scarce. Additionally, we have asked Ashlea Soffrou of Sxan Pro, who we interviewed about software visibility if she could share any images of her software and the effects it produced. We hope that she is able to give a fast response, as we are emailing her late. If she gives us these images, they will be included in the paper in the hope that more revisions are coming our way. We have also mentioned the risk of obsolete medical device donations in our discussion paragraph to raise questions about medical ethics.

Round 2

Reviewer 2 Report

Ms improved. Would be good it you referenced all assertions and insights which have been included as a result of literature searches eg rebound effect, so as to reinforce the power of your text. However the ms is now at a standard which merits publication as it provides a clear analysis of what might be done to make medical care much more sustainable, and identifies barriers to progress.